# Feeding Behavior of *Riptortus pedestris* (Fabricius) on Soybean: Electrical Penetration Graph Analysis and Histological Investigations

**DOI:** 10.3390/insects13060511

**Published:** 2022-05-28

**Authors:** Yan Jin, Wendan Zhang, Yumei Dong, Ai Xia

**Affiliations:** College of Plant Protection, Nanjing Agricultural University, Nanjing 210095, China; 2021802215@stu.njau.edu.cn (Y.J.); 2020802171@stu.njau.edu.cn (W.Z.); 2019202015@njau.edu.cn (Y.D.)

**Keywords:** xylem sap ingestion, cell rupture, EPG, histological examination, “stay green” syndrome

## Abstract

**Simple Summary:**

*Riptortus pedestris* (Fabricius) (Hemiptera: Alydidae) is a major soybean pest with the peak population occurrence during the seed maturity stage from pod filling to harvest. Soybean pods and/or seeds are required for *R. pedestris* development. However, the feeding strategies employed by this stink bug to feed on soybean are still not clear. In the present study, we recorded the feeding behaviors of *R. pedestris* on soybean using electropenetrography (EPG). The biological meaning of each waveform was confirmed by histological examination of plant tissues containing stylets or salivary sheath. In total, five phases of waveforms were identified: non-probing, pathway (Rp1), xylem sap ingestion (Rp2), salivation and ingestion (Rp3), and interruption (Rp4). Xylem ingestion (Rp2) was observed during *R. pedestris* feeding on soybean leaflets, stems, and pods, demonstrating that the stink bug was ingesting xylem sap from vascular tissue. Cell rupture (salivation/ingestion, Rp3) was only recorded during *R. pedestris* feeding on cotyledon and pods. Histological images showed that stylet tips and the salivary sheath were positioned in the tissues of cotyledon and pods. Taken together, our results demonstrate that *R. pedestris* uses a cell-rupture strategy to acquire nutrients from soybean pods and/or seeds while utilizing salivary sheath tactics to obtain water from xylem sap. These findings provide insightful information to understand the interactions between *R. pedestris* and the soybean plant.

**Abstract:**

*Riptortus pedestris* (Fabricius) is a major agricultural pest feeding on soybean pods and seeds. The large populations occur during seed maturity stages from pod filling to harvest. Its infestation results in shriveled and dimpled seeds while vegetative structures (leaflet and stem) remain green, known as “Stay Green” syndrome. Additional evidence also demonstrates that soybean pods and seeds are required for *Riptortus pedestris* development. However, the feeding behavior strategies employed by this stink bug to feed on soybean plants are still not clear. In the present study, the feeding behaviors of *R. pedestris* on soybean plants were recorded by electropenetrography (EPG), and a waveform library was created for this species. A total of five phases of waveforms—nonprobing, pathway (Rp1), xylem sap ingestion (Rp2), salivation and ingestion (Rp3), and interruption (Rp4)—were identified. Non-probing waveforms Z and NP and pathway (Rp1) were found in all tested plant structures (leaflet, stem, cotyledon, and pods). Waveform Rp2 (xylem sap ingestion, xylem ingestion) was primarily recorded during *R. pedestris* feeding on leaflets and stems, while Rp3 (salivation/ingestion) was only observed during feeding on cotyledon and pods. Histological examinations confirmed that correlation between Rp2 and stylet tip positioning in the xylem vessel in leaflets and stems. Stylet tips end in the tissues of cotyledon and pods when Rp3 is recorded. Taken together, our results demonstrate that *R. pedestris* ingests xylem sap from vegetative tissues of soybean (leaflet and stem) via a salivary sheath strategy to obtain water. It mainly acquires nutrients from soybean pods and/or seeds using cell-rupture tactics. This study provided insightful information to understand the field occurrence patterns of “Stay Green” syndrome, which may have important implications for pest control.

## 1. Introduction

*Riptortus pedestris* (Fabricius) (Hemiptera: Alydidae) is a notorious polyphagous agricultural pest. This species has been considered one of the most damaging pests of soybean crops in East Asia, including Korea, Japan, India, and China [1]. Its occurrence has been found on fruit trees, rice, flowering plants, and medical plants, but it prefers leguminous plants, such as soybean (Glycine max), cowpea (Vigna unguiculata), white kidney bean (Phaseolus vulgaris), and mung bean (Vigna radiata) [2,3]. Additional evidence also suggests that only leguminous pods or seeds provide the required nutrients for *R. pedestris* to complete its development and reproduction [2]. Soybean is one of the most important crops, a major source of plant-derived oil and proteins [2]. The large populations of *R. pedestris* in the soybean fields were commonly observed during soybean seed maturity stages from pod filling to harvest in Korea and Japan [4]. This bean bug also poses a serious threat to soybean production in China [5]. The high density of this stink bug occurs when soybean start to bear pods and to harvest from mid-July to late August in the Huang-Huai-Hai River basin, including Henan, Beijing, Shaanxi, and some parts of the Shandong provinces, China [6,7]. Nymphs and adults of *R. pedestris* infest soybean leaves, pods, flowers, and other organs using their needle-like mouthparts, and the typical damage syndrome of injured pods is minute dark spots on the pod walls [8]. Soybean flowers wither away and pods stop growing after suffering *R. pedestris* injury. In soybean fields damaged by *R. pedestris*, pods bear no peas, or contain small and shriveled seeds while leaves and stems remain green during soybean harvest season, known as “Staygreen Syndrome” or “Zhengqing” (Chinese common name) [9,10]. Infestation of *R. pedestris* has caused severe economic loss owing to dramatic soybean yield reduction and low nutritional quality and germination potential of damaged soybean seeds in recent years [7,11]. In south Korea, *R. pedestris* has led to up to 70% of economic losses of soybean production [12]. In China, this species also resulted in tremendous yield losses for soybean growers in the Huang-Huai-Hai River basin [6]. However, what feeding strategies were used by *R. pedestris* to infest soybean plants is still not clear. Why are soybean pods and seeds so important for *R. pedestris* in the soybean field?

Electrical penetration graph or electropenetrography (both names abbreviated EPG) represents an advanced technique that is widely applied to evaluate the feeding behaviors of piercing–sucking insects [13]. EPG allows researchers to analyze various insect stylet activities within plant tissues, including stylet penetration (i.e., inter- versus intracellular), saliva secretion, sap ingestion, and cell rupture [14]. This technique has been used to evaluate the feeding behavior of many herbivorous insects within Hemiptera, such as whitefly, aphid, and psyllids (Stemorrychans) [8,15,16]; leafhoppers and planthoppers (Auchaenorrhynchans) [17]; and stink bugs (Heteropterans) [18]. In 1972, Miles proposed that all hemipterans use one of two different feeding strategies: sheath feeding and lacerate-and-flush feeding [19]. Sheath-feeding insects secrete gelling saliva to form a salivary sheath that surrounds the stylets during its probing in the plant tissue [14]. The function of sheath is thought to anchor the position of the stylets and to lubricate the stylet movements [19]. In lacerate-and-flush feeding, a sheath is not made and stylets move continuously or deeply into the plant tissue to lacerate cells, accompanied by the secretion of watery saliva to “flush out” cell contents; the resulting slurry of cell contents is ingested [14]. More recently, Backus revised the feeding strategies of hemipterans and renamed lacerate-and-flush feeding to the cell-rupture strategy, which has been generally adopted since then [13]. 

Based on many observations in the literature, current investigations revealed that most species of aphids, whiteflies, planthoppers, and leafhoppers use sheath-feeding tactics to ingest xylem and/or phloem sap from plant vascular tissues. Since they acquire nutrients from phloem sap, these species are usually referred to phloem feeders [20,21]. Most of the species within heteropterans use cell-rupture tactics to feed, including the Miridae, such as *Lygus lineolaris* and *Collaria scenica* [22,23], and Pentatomiae species, for example, *Euschistus heros* and *Tibraca limbativentris*, which ingest water from xylem vessels and obtain nutrients using cell-rupture feeding from soybean and rice [18,24]. For the members of Coreidae within Heteroptera, only the feeding behavior of *Anasa tristis* was recorded with EPG so far [25,26], and there is no record of the feeding strategy of *R. pedestris*. 

Given the lack of data available on the feeding behavior of this important soybean pest, the aim of the present study was (1) to investigate *R. pedestris* feeding behavior in soybean plants by creating an EPG waveform library, (2) to determine biological meanings of each electrical waveform using histological investigations and electrical characteristics, and (3) to ascertain feeding sites of *R. pedestris* on the vegetative and reproductive soybean stages. Our findings will elucidate the damage mechanism of *R. pedestris* and contribute to the improved management strategies that will alleviate the economic burden caused by this pest. 

## 2. Materials and Methods

### 2.1. Insect Rearing and Soybean Plants

Adults and nymphs of *R. pedestris* were collected from soybean fields at Shandong and Nanjing, China, in 2019, and since then, have been kept in the laboratory. *R. pedestris* insects were reared in cages (40 cm × 40 cm × 40 cm) at 25 °C under a 16:8 h (light: dark) photoperiod with potted soybean plants and dried seeds as their food source. Soybean (cv. Williams) plants were cultivated at 25 °C and 60% of relative humidity under a 16:8 h (light:dark) photoperiod in plant culture room.

### 2.2. EPG Recordings

The feeding behavior of *R. pedestris* on soybean leaves, pods, and cotyledon was recorded using an eight-channel DC EPG (Giga-8 DC-EPG) system. Before EPG investigation, insects were subjected to hunger treatment for 1 h and then anesthetized with CO_2_. One end of an electrode was adhered to each anesthetized insect thorax with conductive silver glue and then inserted into the EPG probe. Subsequently, a copper-rod plant electrode was inserted into the soil of planted soybean pots to close the electrical circuit for recording. The working principle of EPG was briefly demonstrated in Figure 1A. A total of 50 insects were successfully recorded continually for at least 16 h. The recorded waveform output such as the shape, amplitude, and frequency in hertz was analyzed using software stylet^+^a, which was downloaded from www.epgsystems.eu. Considering that cotyledons are germinated from soybean seeds, waveforms of cotyledon were also described. Due to the small size of the first-to-third-instar nymphs of *R. pedestris* and difficulty of attaching to the gold wire, the fourth- and fifth-instar nymphs and adults (approximately 15–18 days from the first instar) were used to conduct the investigation.

### 2.3. Plant Tissue Histology

To observe stylet pathways of *R. pedestris* in soybean plants, EPG recording was conducted as the previous EPG method. When the specific waveform of interest was observed, the EPG monitor was turned off and the *R. pedestris* stylets were immediately and carefully excised with a sharp micro-scissors. The plant tissues including the leaf, stem, pod, and cotyledon with stylets in were carefully hand-cut into thin sections using a sharp razor blade. The whole process was performed gently to prevent the dislodgement of insect stylets. Some plant tissues containing *R. pedestris* stylets were embedded in paraffin and sectioned with 3–4 μm of thickness as previously described [27]. The dissected plant tissues carrying *R. pedestris* stylets were observed and recorded under an Olympus BX phase contrast microscope (Olympus Corp) using a TOUPCAM digital camera and Imageview software. The salivary sheath was stained using acid fuchsine to facilitate observation. 

## 3. Results

### 3.1. Characterization of EPG Waveforms of R. Pedestris on Soybean Plants

Based on the previous publications of Pentotamidae species [24,28], a family close to Alydidae, and our analysis, a total of six different waveforms were identified for *R. pedestris*—Z, NP, Rp1, Rp2, Rp3 and Rp4—grouped into five different phases—non-probing, pathway (P), ingestion (I), salivation and ingestion, and interruption (N) (Table 1). The types of probing and ingestion and interruption waveforms were labeled by Rp (for *R. pedestris*) followed by a number, as applied in other studies with heteropterans species [24,28].

#### 3.1.1. Noneprobing Waveforms (Z and Np)

During the nonprobing phase, two waveforms were recorded using EPG, and our visual observation confirmed the correlation of these waveforms with stink-bug activities. The Z waveform was observed when the stink bug was standing still on the plant surface (Table 1). The amplitude of the Z waveform was extremely low, showing no visible changes around the baseline at different Ri levels (Figure 1B). The Np waveform consisted of irregular peaks at different Ri levels and was visually correlated with various stink bug body movements such as walking, antennating, or grooming on the plant surface (Figure 2). The Z waves usually interspersed with NP waveforms (Figure 2). 

#### 3.1.2. Probing Waveforms: Pathway Phase–P Family (Rp1) 

The pathway waves occur when the insect stylets penetrate the external and internal plant tissue and secret sheath saliva or rupture pod and cotyledon cells without the secretion of a saliva sheath (Table 1). The RP1 waveform started with voltage levels decreasing abruptly from non-probing waveforms (Z and Np), followed by an irregular pattern (Figure 1 and Figure 2). Rp1 always appeared before Rp2 (xylem sap ingestion; see below) (Figure 1) or Rp3 (salivation and ingestion; see below) (Figure 2). Rp1 was easy to distinguish from Rp2 but sometimes difficult to distinguish from Rp3. 

#### 3.1.3. Xylem Sap Ingestion (Rp2)

The ingestion phase was further divided into Rp2 and Rp3 phases. Among all the waveforms identified, Rp2, representing the activity of *R. pedestris* ingesting of xylem sap, is easy to be recognized by its regular pattern. Rp2 waveform is composed of waves interspersed with regularly distributed downward- or upward-oriented peaks (peaks and waves are illustrated in Figure 3). Amplitude of Rp2 waves and peak orientation vary among Ri level applied.

#### 3.1.4. Salivation and Ingestion (Rp3)

Waveform Rp3 was only observed during feeding activities of *R. pedestris* on soybean cotyledon and pod. Unlike regularly repetitive waveform of Rp2, Rp3 shows irregular patterns and is generally preceded by Rp1 (Figure 2). The detailed waves and peaks of Rp3 are illustrated in Figure 4. Cell-rupture strategy defined by Backus comprised of two different types of waveforms: salivation (cell laceration and maceration of cell tissues) and ingestion (short ingestion of lacerated/macerated cell contents) [13], which is true for *Piezodorus guildinii* [29], *Halyomorpha halys* [28], *Dichelops furcatus* [30], *Euschistus heros* [24], and *Tibraca limbativentris* [18]. However, the waveforms of salivation and ingestion are not clearly distinguished in the case of *R. pedestris*, so we defined salivation/ingestion as Rp3.

#### 3.1.5. Interruption Phase: Saliva Secretion (Rp4)

This family was observed during the long period of ingestion from xylem sap (Rp2) on the leaflets, named the interruption phase (Rp4). Waveform Rp4 shows an irregularly formed flat–spiky plateau dispersed within waveform Rp2, indicating that *R. pedestris* pauses to secrete saliva during the long time of ingestion of xylem sap (Figure 5). 

### 3.2. Correlations between Waveforms and Stylet Tips Position in Plant Tissues

To confirm the biological meaning of each waveform, cross sections of plant tissue containing severed stylets and/or a salivary sheath were examined when a specific waveform of interest was observed. When waveform Rp2 was observed on soybean leaflets, histological examination showed that the severed stylet was located in the tissue of leaflet and ended in the xylem vessels (Figure 6A,B). Figure 6A even revealed that the xylem vessel was damaged after *R. pedestris* stylet penetration. During waveform Rp2, a salivary sheath was also formed surrounding the stylet and finally positioned in the xylem without (Figure 6B,D) or with acid fuchsine staining (Figure 6C). Together, histological investigation ascertained the correlation between waveform Rp2 and xylem sap ingestion in the leaflet. 

We also examined the stylet pathway in the soybean stem using histological investigation and found that when waveform Rp2 appeared, the stylet tip was positioned in the xylem vessel (Figure 6E,F). A salivary sheath was also secreted and wrapped around the stylet during waveform Rp2 (Figure 6E,F). Therefore, histological examinations confirmed the correlation between waveform Rp2 and xylem sap ingestion in soybean stem. 

Waveform Rp3 occurs during *R. pedestris* feeding on the soybean cotyledon and pod. A cross section of cotyledon with stylets showed that stylet tips positioned in the cotyledon (Figure 7A–D). A thin layer of the salivary sheath surrounding stylets was also observed in cotyledon (Figure 7C,D). The histological images during waveform Rp3 in soybean pods showed that *R. pedestris* stylet tips were located within the pod tissue after trespassing the pod wall (Figure 7E). If the pod wall was removed, stylet tips were positioned in the seed endosperm (Figure 7F). Therefore, histological investigation ascertained the correlation between waveform Rp3 and cell-rupture tactics (salivation and ingestion). Rp1 was also observed during *R. pedestris* feeding on soybean pods, but a cross section of plant tissue containing stylet was not achieved because of the short period of this waveform occurrence. We hypothesize that *R. pedestris* might occasionally obtain water from the pod wall. 

## 4. Discussion

The current study is the first paper to study the feeding behavior of a bean bug *R. pedestris* (Heteropterans: Alydidae), an important pest of soybean crops in East Asia. A waveform library was created using the EPG system, and six different waveforms—Np, Z, Rp1, Rp2, RP3 and RP4—were identified associated with the stylet activities of *R. pedestris* on the soybean stem, leaflet, pod, and cotyledon. Waveforms were further grouped into nonprobing, pathway, xylem sap ingestion, salivation and ingestion, and interruption phases (Table 1 and Figure 1, Figure 2 and Figure 5). In comparison with the waveforms recorded from Pentatomidae species, we found that the Rp2 waveform varied in different plant tissues, which is consistent with the DM2 waveforms of *Dichelops melacanthus* on maize seedlings [31] and *Tibraca limbativentris* on Rice [18]. For the cell rupture strategy employed by *R. pedestris*, it is hard to distinguish the waveform salivation from ingestion, so salivation/ingestion was defined together as Rp3, the same as the NV3 of *Nezara viridula* [32]. 

In general, two feeding strategies—salivary sheath and cell rupture—were described for heteropterans [13]. Depending on the species, one or both feeding tactics may be used by stink bugs. The mirid species *Lygus lineolaris* and *Lygus hesperus* use only the cell-rupture strategy when feeding on cotton squares [33]. *Edessa meditabunda* utilizes only salivary-sheath tactics to feed on soybean stems [31]. Most species within Pentatomidae used both the sheath-feeding and cell-rupture strategies, for example, *Piezodorus guildinii* [29], *Euschistus heros* [24], and *Nezara viridula* [32] feeding on soybean plants; *Halyomorpha halys* [28] on vicia faba leaves; *Tibraca limbatinebtris* [18] on rice; and *Dichelops melacanthus* [31] on maize seedlings. We found that *R. pedestris* also used both strategies (salivary sheath and cell rupture) while feeding on soybean plants. Stylet positioning in the xylem vessels during waveform Rp2 demonstrated that *R. pedestris* primarily ingested xylem sap via salivary sheath strategy from soybean leaflets and stems to obtain water. The stylet tips ending in the cotyledon or endosperm of seeds during waveform Rp3 demonstrated that *R. pedestris* ingested cell contents from the cotyledon and pod via cell-rupture tactics. To date, *R. pedestris* is the first reported Alydidae species whose feeding strategy has been investigated using EPG. *Anasa tristis* represented the first Coreidae species whose feeding behavior was studied with EPG in 1999, and this finding only revealed that sheath tips were located in vascular bundles, suggesting that Coreidae insects utilize salivary-sheath tactics to feed; however, more data are needed to further confirm this finding [25].

Stink bugs are generalists and use different plant structures as their most preferred food source. Some stink bugs have a preference for vegetative tissues (leaflet and stem) and acquire nutrients from vascular tissue (phloem) via the salivary-sheath strategy and/or from parenchyma tissue via cell rupturing. For example, *Euschistus meditabunda* and *Tibraca limbativentris* prefer to feed on soybean stems and rice stems [14]. Vegetative structures are usually less nutritious than reproductive structures (pod and seed), which contain essential nutrients including proteins, lipids, and carbohydrates necessary for insect development [14]. So, some stink bugs reported are seed feeders, obtaining nutrients from plant reproductive structures, mostly immature seeds, such as *P. guildinii*, *E. heros*, and *N. viridula*, which prefer to feed on soybean pods [24,29,32]. Our results demonstrate that *R. pedestris* is also a seed feeder, ingesting nutrients from soybean pods and/or seeds via cell rupturing. *Anasa tristis* was the first Coreidae species used to study the feeding behavior with EPG (in 1991), and this finding only revealed that sheath tips were located in vascular bundles [22]. Therefore, our finding in *R. pedestris* represents the most well-studied feeding strategy of Alydidae pests so far.

Previous evidence revealed that only leguminous pods or seeds provided the required nutrients for *R. pedestris* to complete its development and reproduction [2]. A recent study showed that the nymphs of *R. pedestris* completed their individual development and growth successfully when fed with pods while not with leaves, stems, and flowers in vitro [34]. Our finding based on the study of feeding behaviors of *R. pedestris* using EPG analysis and histological investigations provided direct evidence to support the above conclusions because *R. pedestris* only acquires nutrients from soybean pods and seeds via the cell-rupture strategy, while not from vegetative structures (leaflet and stem). In the soybean fields, the most severe period of *R. pedestris* damage was during soybean seed maturity stages from pod filling to harvest in Korea, Japan, and China [4,7]. Our results also provide a reasonable explanation for the occurrence pattern of *R. pedestris* in soybean fields due to the importance of soybean pods to this stink bug’s survival.

## 5. Conclusions

In conclusion, our results demonstrate that *R. pedestris* feeds on soybean plants in two different ways: (1) ingestion of cell contents from cotyledons and seed endosperm of pods via cell-rupture strategy to obtain nutrients, and (2) ingestion of water from xylem sap of the leaflets and stems using salivary-sheath strategy. Our results support the previous investigations stating that the nutrients of soybean pods or seeds are required for *R. pedestris* to complete its development and reproduction. Our finding also provides a reasonable explanation for the prevalence of stay-green syndrome during soybean seed maturity stages from pod filling to harvest. 

## Figures and Tables

**Figure 1 insects-13-00511-f001:**
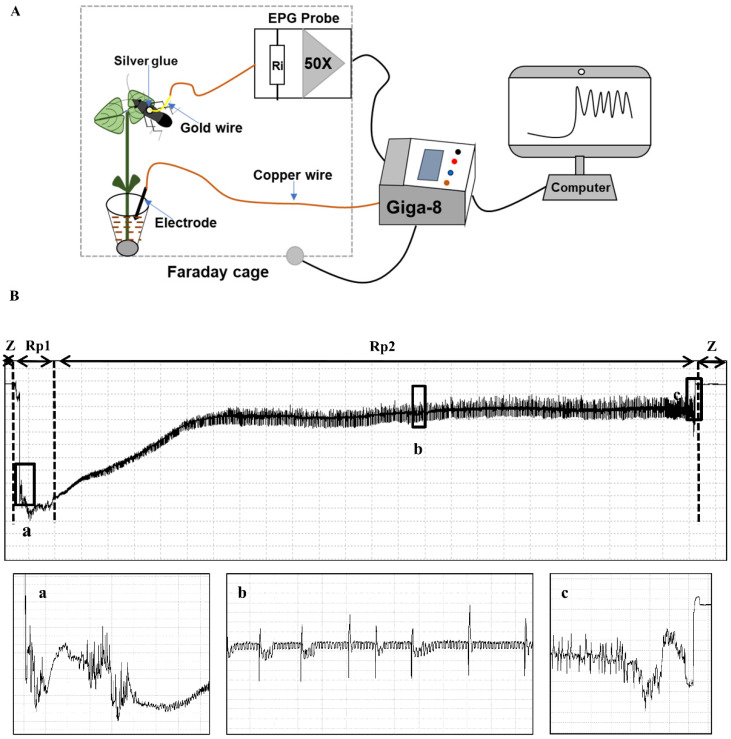
Demonstration of working principle of EPG (**A**), and waveforms generated using EPG for *R. pedestris* on leaflet, stem, and pod of soybean (**B**). *Box a* indicated an enlarged view of beginning of probing pathway (Rp1) from Figure 1B. *Box b* was an enlarged view of a part of waveform Rp2 indicated from Figure 1B. *Box c* showed an enlarged view of the end of Rp2 waveform when insect retracts its stylets from plant tissue on Figure 1B.

**Figure 2 insects-13-00511-f002:**
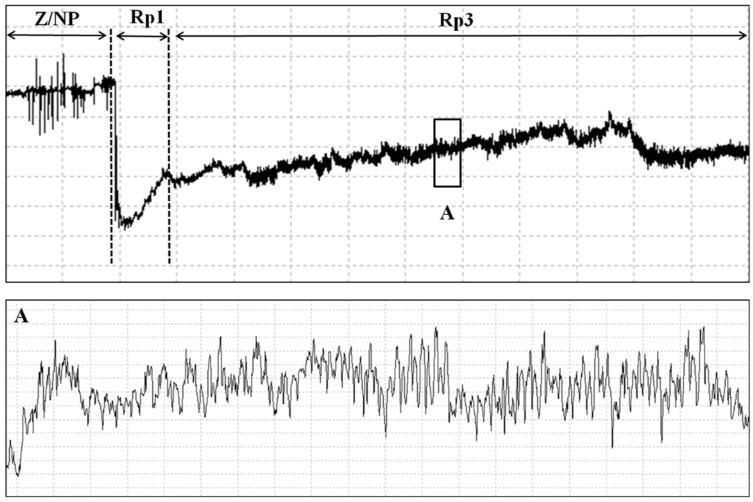
Waveforms generated using EPG for *R. pedestris* on pod and cotyledon of soybean. *Box A* enlarged view of detail of waveform Rp3.

**Figure 3 insects-13-00511-f003:**
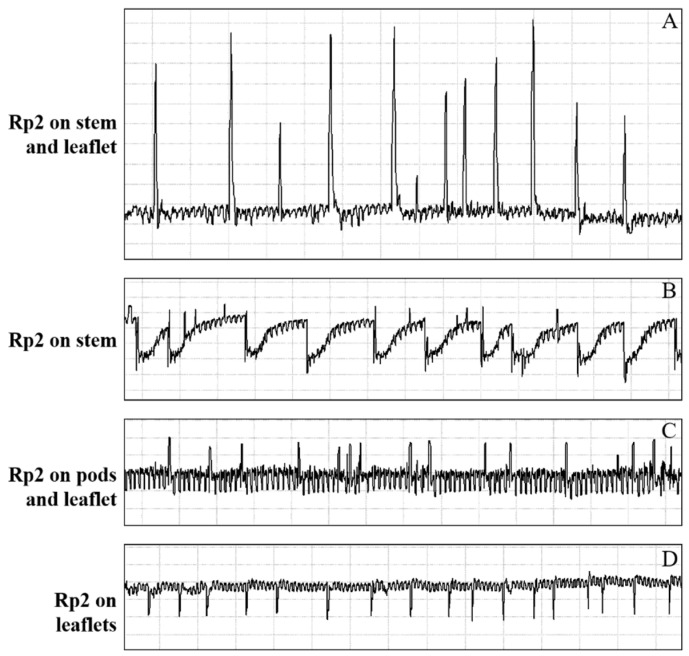
Detail of the waveform Rp2 recorded during feeding behavior of *R. pedestris* on soybean leaflet, stem, and pod. (**A**) Amplitude of Rp2 waves and peak orientation recorded on stem and leaflet with relative amplitude (−4.3–−2.3). (**B**) Amplitude of Rp2 waves and peak orientation recorded on stem with relative amplitude (−0.2–0.3). (**C**) Amplitude of Rp2 waves and peak orientation recorded on pod and leaflet with relative amplitude (−0.6–0.3). (**D**) Amplitude of Rp2 waves and peak orientation recorded on leaflet with relative amplitude (−3.5–−3.3).

**Figure 4 insects-13-00511-f004:**
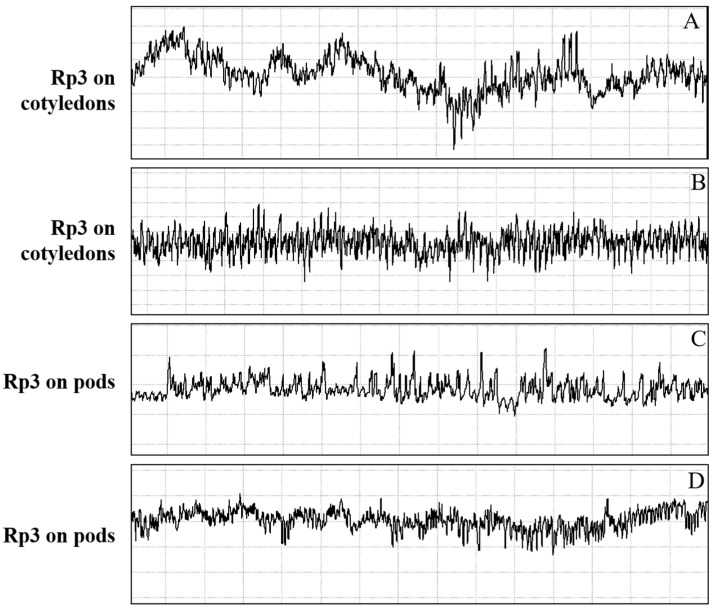
Detail of the waveform Rp3 recorded during feeding behavior of *R. pedestris* on soybean cotyledon and pod. (**A**) Amplitude of Rp3 waves and peak orientation recorded on cotyledons with relative amplitude (−3.8–−2.4). (**B**) Amplitude of Rp3 waves and peak orientation recorded on cotyledons with relative amplitude (−2.4–−1.8). (**C**) Amplitude of Rp3 waves and peak orientation recorded on pods with relative amplitude (−4.5–−3.5). (**D**) Amplitude of Rp3 waves and peak orientation recorded on pods with relative amplitude (0.8–1.2).

**Figure 5 insects-13-00511-f005:**
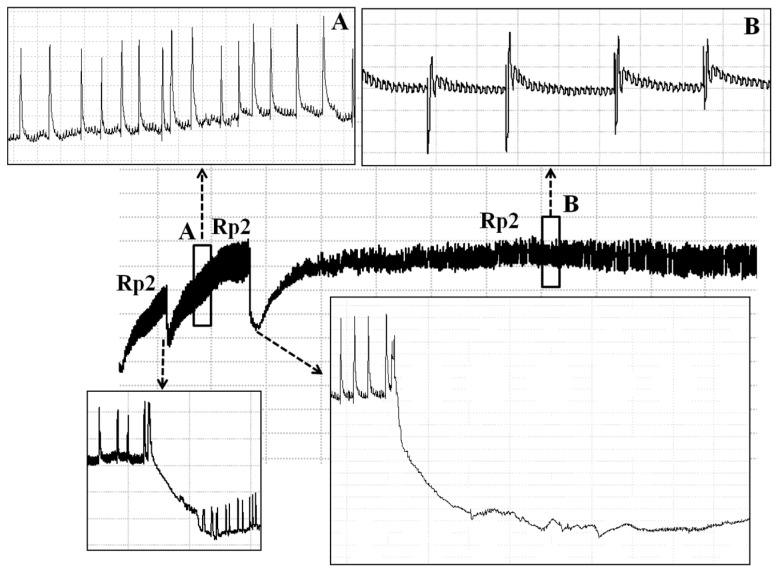
Detail of the interruptions (waveform Rp4) observed during waveform Rp2. *Box A* and *B* enlarged view of a regular pattern of waveform Rp2. Black dashed line represents the enlarged views of waveform interruption (Rp4).

**Figure 6 insects-13-00511-f006:**
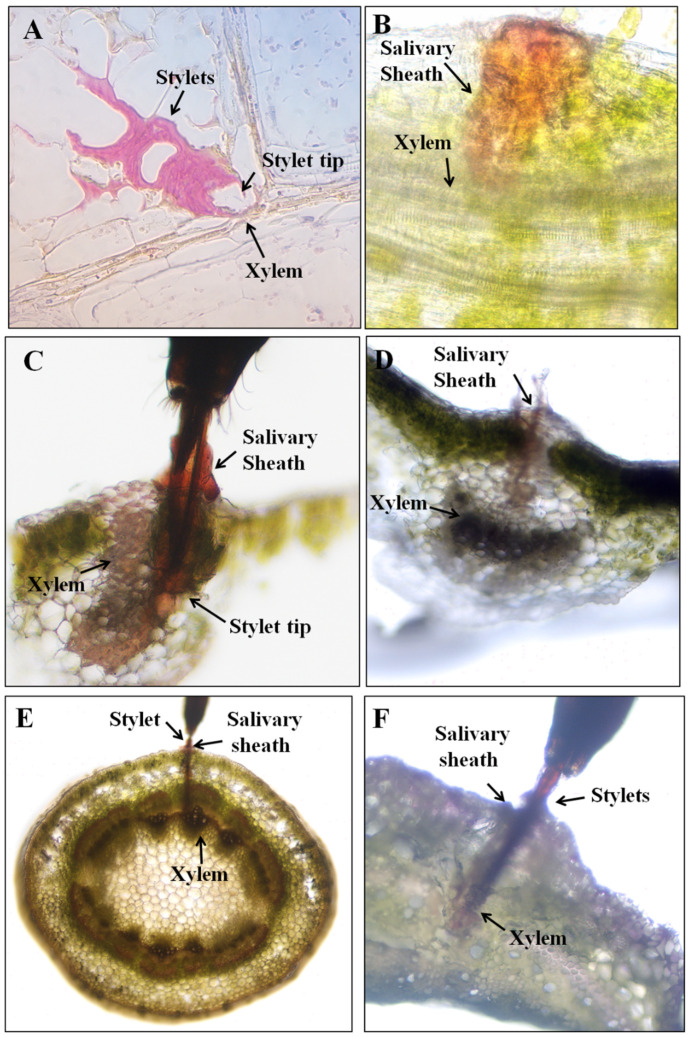
Cross sections of soybean leaflets and stems containing severed stylets and/or salivary sheath of *R. pedestris* during waveform Rp2. (**A**) Complete stylet tips ending in the xylem vessels of leaflet during waveform Rp2 (Paraffin section). (**B**,**D**) Complete salivary sheath ending in the xylem vessels of leaflet during waveform Rp2. (**C**,**D**). Complete stylet tips ending in the xylem vessels of leaflet during waveform Rp2. Salivary sheath was stained with acid fuchsine. (**E**) Salivary sheath stained with acid fuchsine in the stem tissue. (**F**) Stylets and secreted salivary sheath ending in the xylem vessel.

**Figure 7 insects-13-00511-f007:**
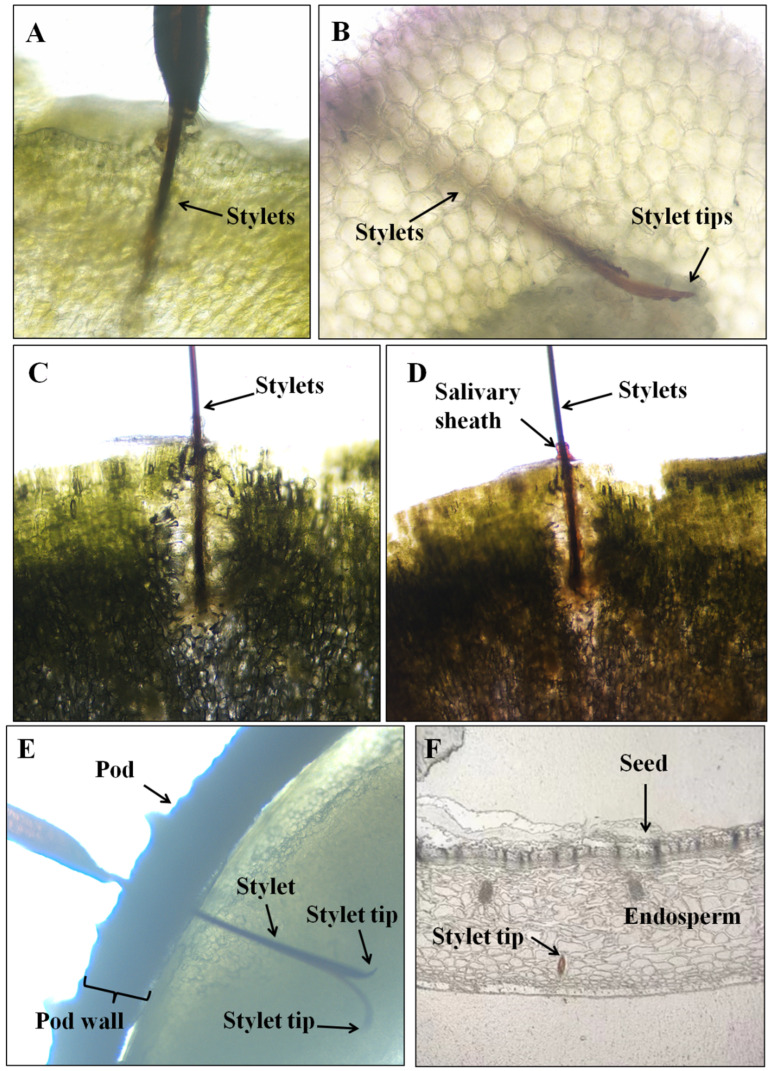
Cross sections of soybean cotyledons and pods containing severed stylets of *R. pedestris* during waveform Rp3. (**A**–**C**). Complete stylet tips ending in the cell tissue of cotyledon during waveform Rp3. (**D**). Complete salivary sheath stained with acid fuchsine ending in the cell tissue of cotyledon during waveform Rp3. (**E**). Complete stylet tips ending in the cell tissue of pod during waveform Rp3. (**F**). Complete stylet tips ending in the seed during waveform Rp3 (Paraffin section).

**Table 1 insects-13-00511-t001:** Description of waveforms recorded using EPG during the feeding of *Riptortus pedestris* instars and adults on soybean plants, and their biological meanings.

Phase	Family	Type or Subtype	Plant Structure	Relative Amplitude (%)	Frequency (Hz)	Biological Meaning
Nonprobing		Z	Leaflet/stem/pod/cotyledon	Very low	Irregular	Stand still on plant surface
		NP	Leaflet/stem/pod/cotyledon	Low/high	Irregular	Various body movements (Walking, antennating or grooming) on the plant surface
Pathway	P	Rp1	Leaflet/stem/pod/cotyledon	83 (67–100)	Irregular	Stylet penetration with salivary sheath secretion
Ingestion	I	Rp2	Leaflet/stem/pod	45 (15–83)	3.75 Hz (3.0–4.5)	Xylem sap ingestion
Salivation and ingestion	I	Rp3	Pod/cotyledon	70 (30–77)	Irregular	Watery saliva secretion, cell laceration and enzymatic maceration, and ingestion of lacerated/macerated cell tissues
Interruption	N	Rp4	leaflet		Irregular	Brief interludes of salivation during Xylem sap ingestion

## Data Availability

Data available in a publicly accessible repository.

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
