# Peer review of "Feeding Behavior of Riptortus pedestris (Fabricius) on Soybean: Electrical Penetration Graph Analysis and Histological Investigations"

_insects, 2022, doi:10.3390/insects13060511_

Round 1

Reviewer 1 Report

- Consider to rewrite the title. see comment on article

- Line 43, check tense

-Lines 73-87, check citations for consistency

-Lines 92, check tense

-Lines 109-110, rewrite to enhance flow of methods

-Section 3.1 which is under results is actually methods. consider to move to methods section

-Lines 190, Figure 4 caption should be improved to improve on clarity

-see other minute comments in the article.

Author Response

Dear anonymous reviewer,

Here, we submit the revised manuscript “Feeding behavior of Riptortus pedestris (Fabricius) on soybean: Electrical penetration graph analysis and histological investigations” for your consideration. We thank you for your helpful comments. In this letter, we addressed all the points raised you and included a detailed list (point-to-point) of responses to the comments. All the changes we have made in the manuscript was marked with track changes.

Yours sincerely,

Ai Xia

--------------------------------------------------------------------------------------------

Responses to Comments and Suggestions for Authors

- Line 43, check tense

Response: Corrected and changed into “This bean bug also poses a serious threat to soybean production in China [4].” Please see revised manuscript (Line 58-59).

-Lines 73-87, check citations for consistency

Response: Citations in the whole manuscript has been checked and corrected.

-Lines 92, check tense

Response: Corrected and changed into “the aim of the present study was …”. Please see revised manuscript (Line 106).

-Lines 109-110, rewrite to enhance flow of methods

Response: Rewrote. “Before EPG investigation, insects were subjected to hunger treatment for 1h and then anesthetized with CO2. One end of electrode was adhered to each anesthetized insect thorax with conductive silver glue….”. Please see revised manuscript (Line 122-124).

-Section 3.1 which is under results is actually methods. consider to move to methods section

Response: I agree with you, but after consideration, partial results were moved to the section 2.2. EPG recording, please see revised manuscript (Line 127-134). But the rest of results should be kept in the 3.1, please see revised manuscript (Line 149-154).

-Lines 190, Figure 4 caption should be improved to improve on clarity

Response: Figure 4 even Figure 3 captions were improved, please see revised manuscript.

-see other minute comments in the article.

Other Comments in the article:

Title: the part starting “a pest responsible for staygreen syndrome on soybean plants” could be deleted.

Response: I agree to delete “a pest responsible for staygreen syndrome” but still keep “soybean”. So the title is “Feeding behavior of Riptortus pedestris (Fabricius) on soybean: Electrical penetration graph analysis and histological investigations”.

In text "Fig." is used. consider to either use "Figure" or "Fig" and be consistent.

Response: All the figures in the text were changed into “Figure X” to make consistency with Figure captions.

Line 199 Figure 5: Red dash line is not visible.

Response: Changed into black dash line.

Reviewer 2 Report

This manuscript reports an interesting study of behavioral strategies of host-plant by Riptortus pedestris (Fabricius) using EPG analysis and histological investigations. The authors conclude that R. pedestris ingests xylem sap from vegetative tissues of soybean (leaflet and stem) via salivary sheath strategy to obtain water. This finding has the potential to explain field occurrences patterns of “stay green” syndrome. The results are properly documented, I have just minor suggestions for improvement.

  1. L43, is it a citation?
  2. L97, I suggest replacing “insect” with “pest”.
  3. L61, this description of the EPG is lacking much understanding about its working principle for non-professionals.
  4. L79-81, several references are missing here.
  5. L81, 84,85,87, please check the citation format through the whole manuscript.
  6. L115, the software version as well as parameters used are missing from the description.
  7. L135, it is unclear which day of the 4th and 5th instar nymphs?
  8. L136, what basis are these six different waveforms identified?
  9. L173, Fig. 3)?
  10. L206, Fig.A?
  11. In the discussion section, the authors could add a few sentences on what their findings might mean in general for pedestris feeds on soybean plant in two different ways. Do other Coreidae insects also require “two diiferent ways.”

Reviewer 3 Report

The manuscript entitled: Feeding behavior of Riptortus pedestris (Fabricius),a pest responsible for staygreen syndrome, on soybean plants: EPG analysis and histological
investigations, by Yan Jin , Wendan Zhang , Yumei Dong , Ai Xia aims to
investigate the feeding behavior of R. pedestris in Soy.
The manuscript is interesting and the subject and scope are appropriate for the journal. However, I have  some concerns that need  to be addressed.
I emphasize some important points to be observed:
The term EPG used in the title should be replaced by Electrical penetration graph and so EPG can be maintained in keywords;
In keywords, Riptortus pedestris; feeding behavior and soybean plants should be replaced;
Introduction - line 22 add (Hemiptera: Alydidae);
Lines 22 and 23 delete "is classified into the class of Coreidae, suborder
of Heteropterans and order of Hemiptera";
Material and methods - item 2.2 - adding a scheme or image to show the methodology, would be interesting;
Item 2.3. line 125 - what size of section?
In material and methods, comment that the description of the waveforms
recorded by the EPG will be done. Define which “family” which is described in table 1;
Line 132 - in item 2.1 - you describe that you will analyze;
Non-Probing ítem 3.1.1 or Nonprobing in table 1;
Line 18, 20, 162, 168 - The term xylem ingestion is not appropriate throughout the text, replace it with xylem sap ingestion because what is ingested is the sap;
Line 173 the 175 delete (discussion);
Saliva injection (line 192) or secrete saliva (line 196);
Line 193 ingestion of xylem sap and not cells;
In item 3.1.3.e 3.1.4 observe that the authors mix results with discussion, rewrite!
I hope these comments are useful.
